# Risk of recurrence and bleeding in patients with cancer-associated venous thromboembolism in the direct oral anticoagulants era: Findings from the TULIPE registry

Yuki Ishizuka[1,2,3], Kazuko Tajiri [3,4]*, Hiroyuki Naito[3,5], Momoko Murata[6], Masayuki Hattori[2], Tomoko Machino-Ohtsuka[2], Tomoko Ishizu[2]

1 Department of Cardiology, Tokyo Metropolitan Bokutoh Hospital, Tokyo, Japan, 2 Department of Cardiology, Institute of Medicine, University of Tsukuba, Tsukuba, Japan, 3 Department of Cardiology, National Cancer Center Hospital East, Kashiwa, Japan, 4 Tsukuba Life Science Innovation Program (T-LSI), School of Integrative and Global Majors (SIGMA), University of Tsukuba, Tsukuba, Japan, 5 Clinical Laboratory, Hidaka Hospital, Takasaki, Japan, 6 Department of Clinical Laboratories, National Cancer Center Hospital East, Kashiwa, Japan

* ktajiri@east.ncc.go.jp

## Abstract

### Background

The introduction of direct oral anticoagulants (DOACs) for venous thromboembolism (VTE) treatment has led to their widespread adoption in clinical practice, potentially influencing management strategies and patient outcomes. However, real-world data on cancer-associated VTE in the DOAC era remain limited. This study aimed to evaluate the clinical characteristics and long-term outcomes of patients with cancer- and non-cancer-associated VTE in a real-world setting.

### Methods

We retrospectively analyzed patients diagnosed with deep vein thrombosis (DVT) using lower-extremity venous ultrasound between January 2015 and August 2020 at the University of Tsukuba Hospital, a tertiary academic referral center in Japan.

### Results

The cohort included 2,281 patients with DVT, comprising 1,152 with active cancer (cancer group) and 1,129 without cancer (non-cancer group). The cumulative 5-year incidence of recurrent VTE was significantly higher in the cancer group than in the non-cancer group (25% vs. 10%, P < 0.001). After adjusting for confounders and accounting for the competing risk of mortality, cancer remained a significant risk factor for recurrence (adjusted subdistribution hazard ratio [sHR]: 2.00; 95% confidence interval [CI]: 1.46–2.74). Similarly, the cumulative 5-year incidence of major bleeding

**Data availability statement:** The data underlying the results presented in the study are available from the corresponding author on reasonable request. There are ethical restrictions on sharing a de-identified dataset because the data contain potentially identifying or sensitive patient information, imposed by the Institutional Review Board of the University of Tsukuba Hospital. Institutional Review Board of the University of Tsukuba Hospital Amakubo 2-1-1, Tsukuba, Ibaraki 305-8576 rinshoken-kyu@un.tsukuba.ac.jp.

**Funding:** This work was funded by the National Cancer Center Research and Development Fund (2023-A-12) and the MHLW Research for Promotion of Cancer Control Programs (grant number 23EA1036) to KT. The funders did not play any role in the study design, data collection and analysis, decision to publish, or preparation of the manuscript.

**Competing interests:** I have read the journal's policy and the authors of this manuscript have the following competing interests: KT has received honoraria from Bristol Myers Squibb, Pfizer, Bayer, and Daiichi Sankyo. The remaining authors declare that the research was conducted in the absence of any commercial or financial relationships that could be construed as a potential conflict of interest.

was significantly higher in the cancer group (30% vs. 9.6%, P < 0.001). After adjustment, the risk of major bleeding remained significantly elevated in the cancer group compared to that in the non-cancer group (adjusted sHR: 2.69; 95% CI: 1.90–3.81). In the cancer group, discontinuation of bleeding-related anticoagulation therapy was associated with increased VTE recurrence (P < 0.001), whereas no such association was observed in the non-cancer group (P = 0.716).

## Conclusions

In the DOAC era, similar to the warfarin era, patients with cancer exhibited significantly higher rates of VTE recurrence and major bleeding than those without cancer.

## Introduction

Patients with cancer are at a significantly higher risk of developing thrombosis than those without cancer, a condition known as cancer-associated thrombosis (CAT). The incidence of venous thromboembolism (VTE) has increased in parallel with cancer survival rates, largely because of advancements in cancer treatment [1,2].

Managing cancer-associated VTE presents significant clinical challenges, including elevated bleeding risk, patient frailty, and potential drug–drug interactions. Recent guidelines recommend direct oral anticoagulants (DOACs) as the first-line anticoagulant therapy for CAT, except in cases involving luminal gastrointestinal and genitourinary cancers [3–5]. However, these recommendations are primarily based on randomized clinical trials (RCTs) conducted in highly selected patient populations under controlled conditions, and may not accurately represent real-world clinical practice. Although a recent large-scale observational study from the DOAC era found that patients with CAT had a higher risk of major bleeding but no increased long-term risk of VTE recurrence after adjusting for the competing risk of death [6], real-world data on CAT in the DOAC era remain limited. In particular, existing studies, such as the COMMAND VTE Registry and COMMAND VTE Registry-2, primarily focused on symptomatic VTE and did not include asymptomatic DVT [6,7], which is increasingly detected through routine ultrasonography in patients with cancer. However, previous studies have shown that asymptomatic DVT is associated with poor survival in ambulatory patients with cancer [8], and even asymptomatic isolated distal DVT carries a non-negligible risk of recurrence, especially after discontinuation of anticoagulation therapy [9]. These findings underscore the clinical importance of asymptomatic DVT in the field of oncology. Therefore, it is essential to evaluate both symptomatic and asymptomatic DVT in real-world practice to better understand its prognostic implications and inform anticoagulation decisions in patients with cancer.

This study aimed to evaluate the clinical characteristics and long-term outcomes of patients with cancer- and non-cancer-associated VTE in real-world settings, including both symptomatic and asymptomatic patients. The primary outcomes of this study were the incidence of VTE recurrence, major bleeding, and overall survival, each of which was compared between the active cancer and non-cancer groups. Secondary

outcomes included the impact of anticoagulation discontinuation, particularly because of bleeding, VTE recurrence, major bleeding, overall survival, and cancer-type-specific risks of VTE recurrence and major bleeding. These were evaluated using multivariable Fine–Gray and Cox proportional hazards models, as appropriate.

## Materials and methods

### Study design

The Tsukuba Ultrasound for Lower-Extremity Deep Vein Thrombosis (TULIPE) registry is a single-center, retrospective cohort study that included consecutive patients who underwent lower-extremity venous ultrasound between September 1, 2011, and August 31, 2020, at the University of Tsukuba Hospital, a tertiary academic referral center in Japan. To facilitate a structured analysis, data extracted from the TULIPE registry were systematically categorized into predefined domains, including demographics, clinical characteristics, laboratory investigations, treatment information (e.g., anticoagulant type and duration), and clinical outcomes (e.g., VTE recurrence, major bleeding, and overall survival). Demographic data included age, sex, and body mass index (BMI). The clinical characteristics included comorbidities (e.g., hypertension and diabetes), type and location of DVT (proximal or distal), presence of pulmonary embolism (PE), cancer type, and metastatic status. Laboratory findings included hemoglobin, platelet count, D-dimer, and C-reactive protein (CRP) levels, which were selected based on prior evidence of their relevance to VTE recurrence and bleeding risk [10,11]. Treatment-related variables included the type of anticoagulant used (e.g., DOACs or warfarin), duration of therapy, and reasons for discontinuation. Anticoagulation therapy beyond the acute phase was defined as continuation beyond 10 days after VTE diagnosis, in accordance with the literature [6]. The study was conducted in accordance with the principles outlined in the Declaration of Helsinki and was approved by the Institutional Review Board of the University of Tsukuba Hospital (approval number: R02-183). The requirement for informed consent was waived owing to the retrospective nature of the study. We did not have access to information that could identify the individual participants during or after data collection. We accessed the data for research purposes on January 7, 2021, and October 2, 2022.

### Study population

A total of 17,188 lower-extremity venous ultrasound examinations were performed in the TULIPE registry, involving 10,303 patients. As this study aimed to investigate VTE recurrence and bleeding events in the DOAC era, 4,234 examinations were excluded because the ultrasound was performed before January 1, 2015. Additional exclusion criteria were patients without DVT, those undergoing ultrasonography for DVT follow-up, and those evaluated for lower-extremity varicose veins. The final study cohort comprised 2,281 patients who were divided into two groups based on the presence or absence of cancer at the time of VTE diagnosis: 1,152 patients with active cancer (cancer group) and 1,129 patients without active cancer (non-cancer group) (Fig 1). Our registry did not impose an age limit on patient registration; patients aged 3–101 years were enrolled. However, only patients aged 20–98 years met the inclusion criteria. Consequently, all the eligible patients were adults. The cancer group was defined as patients with a cancer diagnosis within 6 months prior to ultrasound; recurrent, regionally advanced, or metastatic cancer; cancer treatment within the previous 6 months; or hematologic cancer not in complete remission (as previously described) [12].

### Clinical outcomes

The primary outcomes were recurrent VTE, major bleeding, and overall survival. Recurrent VTE was defined as new-onset lower-extremity venous thrombosis, PE, or Trousseau syndrome confirmed by imaging (e.g., venous ultrasound or contrast-enhanced computed tomography [CT]), regardless of symptoms, based on previously published criteria [7]. Major bleeding was defined according to the International Society on Thrombosis and Hemostasis [7,13,14], including hemoglobin loss >2 g/dL, transfusion of more than 2 units of blood, bleeding in critical sites (intracranial, intraspinal, intraocular,

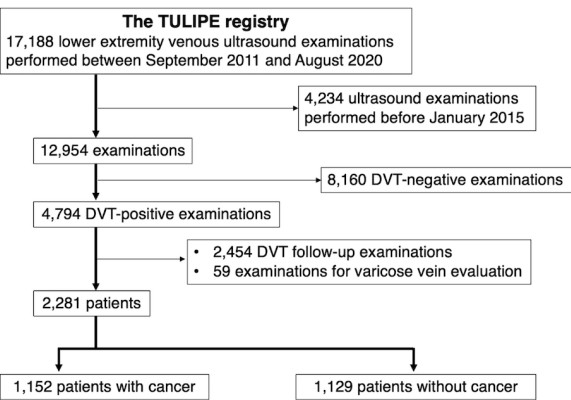

**Fig 1. Study flow chart.** DVT, deep vein thrombosis.

pericardial, intraarticular, intramuscular with compartment syndrome, and retroperitoneal), and bleeding that required surgical intervention or led to death. Discontinuation of anticoagulation therapy was defined as an interruption of therapy for > 14 days. Overall survival (OS) was defined as the number of days from the DVT diagnosis to death from any cause. For patients who were transferred to other hospitals or received home-based care, we attempted to collect outcome data by obtaining referral letters or discharge summaries from the treatment institutions.

## Statistical analysis

Continuous variables were assessed for normal distribution using the Shapiro–Wilk test and are presented as mean ± standard deviation for normally distributed data or median (interquartile range [IQR]) for non-normally distributed data. Comparisons were made using one-way analysis of variance or Kruskal–Wallis test, as appropriate. Categorical variables are presented as numbers or percentages, and comparisons were performed using the chi-square test or Fisher's exact test, as appropriate. The Kaplan–Meier method was used to estimate the cumulative incidence, and differences between groups were evaluated using the log-rank test. The Cox proportional hazards model was used to estimate the hazard ratios (HRs) and corresponding 95% confidence intervals (CIs) for patients with cancer compared to those without. Given the potential impact of mortality as a competing risk, the Fine–Gray subdistribution hazard model was applied to estimate the subdistribution hazard ratios (sHRs) for VTE recurrence and major bleeding. In these models, death from other causes was treated as a competing event, whereas in the Cox model, it was treated as a censored event for comparison. Multivariate models were adjusted for age, sex, BMI, comorbidities as a whole, baseline PE, hemoglobin level, anticoagulation beyond the acute phase, and discontinuation of anticoagulant therapy. Notably, discontinuation was excluded from the adjustment in the major bleeding model because bleeding events occurring after discontinuation were not systematically captured in our registry. Subgroup analyses for VTE recurrence and major bleeding were conducted using Fine–Gray subdistribution hazard models, adjusting for the same covariates in the primary analyses. To evaluate the impact of VTE recurrence and major bleeding on OS, additional subgroup analyses were performed using Cox proportional hazards models, incorporating VTE recurrence or major bleeding as covariates along with the previously described variables. All statistical analyses were conducted using R version 4.3.2 (R Foundation for Statistical Computing, Vienna, Austria; www.r-project.org). All P-values were two-tailed, and statistical significance was defined as $P < 0.05$. The proportion of missing data was very low (<5%) across all assessed variables: BMI (1.4%), hemoglobin level (0.4%), platelet count (0.5%), D-dimer level (1.0%), and CRP level (0.7%). To address these missing values, multiple imputations were performed using chained equations (MICE) to generate five imputed datasets.

## Results

### Baseline characteristics

**Demographic data.** The patient characteristics are summarized in Table 1. Several significant differences were observed between the cancer and non-cancer groups. Patients in the cancer group were younger (71 years vs. 72 years, P=0.018) and had a lower BMI (22.0 kg/m$^2$ vs. 22.8 kg/m$^2$, P<0.001).

**Clinical data.** The cancer group had lower hemoglobin levels (10.7 g/dL vs. 11.1 g/dL, P<0.001), and higher CRP levels (1.0 mg/dL vs. 0.8 mg/dL, P=0.002) compared with the non-cancer group. The prevalence of comorbidities, including hypertension (45.7% vs. 55.4%, P<0.001), dyslipidemia (25.7% vs. 33.1%, P<0.001), diabetes mellitus (19.0% vs. 25.6%, P<0.001), and history of VTE (4.1% vs. 8.0%, P<0.001), was lower in the cancer group. The incidence of PE was higher in the cancer group than in the non-cancer group (10.9% vs. 6.0%, P<0.001). However, PE severity was lower in the cancer group, as indicated by reduced rates of cardiac arrest or collapse (8.7% vs. 35.3%, P<0.001), shock, and hypoxemia (19.0% vs. 32.4%, P=0.038). No significant difference was observed in the DVT location between the two groups, with distal DVT being the most common presentation.

Table 2 shows the characteristics of the patients in the cancer group. Cervical or uterine cancer was the most frequently observed malignancy (15.5%), followed by ovarian or peritoneal cancer (10.8%); kidney, ureter, or bladder cancer (9.7%); central nervous system (CNS) cancer (9.5%); and lung cancer (9.5%). Among the patients with cancer, 33.9% had distant metastases.

**Treatment strategies.** Anticoagulation strategies varied significantly between the two groups (Table 3). Initial parenteral anticoagulation therapy was administered more frequently in the cancer group than in the non-cancer group (26.0% vs. 21.9%, P=0.023). Anticoagulation therapy beyond the acute phase was also more frequently implemented in patients with cancer than in those without (78.8% vs. 70.6%, P<0.001). Warfarin use was more frequent in the non-cancer group (7.3% vs. 18.7%, P<0.001), whereas DOACs were more commonly prescribed in the cancer group (86.6% vs. 75.8%, P<0.001). The trends in anticoagulant use during the study period are presented in S1 Fig. In both the cancer and non-cancer groups, the percentage of DOAC prescriptions increased over time, whereas the percentage of prescriptions for warfarin and other anticoagulants decreased. Among patients receiving anticoagulation therapy beyond the acute phase, the cumulative discontinuation rates were similar between the two groups (P=0.444) (S2 Fig). However, discontinuation because of bleeding events was significantly more frequent in the cancer group (28.5% vs. 13.9%, P<0.001) (Table 3). There was no significant difference in the annual rate of anticoagulation therapy discontinuation during the study period (S3 Fig).

### Clinical outcomes

**VTE recurrence.** During a median follow-up of 594 days (IQR: 117–1,125), VTE recurrence occurred in 132 (11.5%) and 60 (5.3%) patients in the cancer and non-cancer groups, respectively (P<0.001) (Table 4). As illustrated in Fig 2, the cumulative incidence of recurrent VTE was significantly higher in the cancer group at 1 (8.6% vs. 3.7%), 3 (18% vs. 6.7%), and 5 years (25% vs. 10%) (log-rank P<0.001). A similar trend was observed in the analysis restricted to patients who received anticoagulation therapy (S4 Fig). We further stratified the patients into three groups—non-cancer, cancer without distant metastasis, and cancer with distant metastasis—to explore the differences in clinical outcomes. We found that VTE recurrence was lowest in the non-cancer group, with no significant differences between the cancer groups with and without metastasis (S5 Fig). We evaluated the year-by-year trends in VTE recurrence (S6 Fig). In the cancer group, the cumulative incidence of VTE recurrence varied significantly by year, with notably higher recurrence rates observed in the latter part of the study period (2019–2020). In contrast, no significant differences in VTE recurrence by year were observed in the non-cancer group. To explore the factors contributing to these annual differences within the cancer group, we conducted Kaplan–Meier analyses, stratifying cases into symptomatic and asymptomatic VTE recurrences

**Table 1. Patient characteristics.**

| Characteristics | Active cancer group (n = 1,152) | Non-cancer group (n = 1,129) | P-value |
|---|---|---|---|
| **Age (years)** | 71.0 (63.0–77.0) | 72.0 (63.0–79.0) | 0.018 |
| **Women** | 750 (65.1%) | 758 (67.1%) | 0.305 |
| **Body mass index[a] (kg/m$^2$)** | 22.0 (19.3–24.6) | 22.8 (20.0–25.5) | <0.001 |
| **Comorbidities** | 866 (75.2%) | 1,103 (97.8%) | <0.001 |
| **Hypertension** | 526 (45.7%) | 626 (55.4%) | <0.001 |
| **Dyslipidemia** | 296 (25.7%) | 374 (33.1%) | <0.001 |
| **Diabetes mellitus** | 219 (19.0%) | 289 (25.6%) | <0.001 |
| **Chronic kidney disease** | 334 (29.0%) | 358 (31.7%) | 0.158 |
| **Chronic lung disease** | 61 (5.3%) | 95 (8.4%) | 0.003 |
| **Heart failure** | 79 (6.9%) | 210 (18.6%) | <0.001 |
| **Autoimmune disease** | 76 (6.6%) | 313 (27.7%) | <0.001 |
| **Orthopedic disease** | 61 (5.3%) | 315 (27.9%) | <0.001 |
| **Neurological disease** | 60 (5.2%) | 293 (26.0%) | <0.001 |
| **Varicose vein** | 10 (0.9%) | 29 (2.6%) | 0.002 |
| **Mental illness** | 39 (3.4%) | 116 (10.3%) | <0.001 |
| **Pregnancy/puerperium** | 0 (0%) | 19 (1.7%) | <0.001 |
| **Liver cirrhosis** | 7 (0.6%) | 9 (0.8%) | 0.588 |
| **History of VTE** | 47 (4.1%) | 90 (8.0%) | <0.001 |
| **Presentation** | | | |
| **DVT** | | | |
| **Proximal type** | 218 (18.9%) | 207 (18.3%) | 0.718 |
| **Distal type** | 934 (80.2%) | 922 (81.6%) | |
| **PE** | 126 (10.9%) | 68 (6.0%) | <0.001 |
| **Symptomatic** | 26/126 (20.6%) | 24/68 (35.3%) | 0.026 |
| **Shock/low blood pressure** | 3/126 (2.4%) | 2/68 (2.9%) | >0.999 |
| **Hypoxemia** | 24/126 (19.0%) | 22/68 (32.4%) | 0.038 |
| **Cardiac arrest, collapse** | 11/126 (8.7%) | 24/68 (35.3%) | <0.001 |
| **Laboratory tests at diagnosis** | | | |
| **Hemoglobin[b] (g/dL)** | 10.7 (9.3–12.3) | 11.1 (9.8–12.7) | <0.001 |
| **Platelet[c] (×10$^4$/μL)** | 23.0 (16.9–30.8) | 23.1 (18.0–29.7) | 0.234 |
| **D-dimer[d] (μg/mL)** | 4.3 (1.9–9.2) | 5.0 (2.0–11.0) | 0.195 |
| **C-reactive protein[e] (mg/dL)** | 1.0 (0.1–4.1) | 0.8 (0.1–3.2) | 0.002 |

Data are expressed as n (%) or as median (interquartile range).

[a]Body mass index had 33 missing values.

[b]Hemoglobin had 10 missing values.

[c]Platelet had 11 missing values.

[d]D-dimer had 22 missing values.

[e]C-reactive protein had 17 missing values.

DVT, deep vein thrombosis; PE, pulmonary embolism; VTE, venous thromboembolism.

(S7 Fig). The results showed no significant variation by year in symptomatic VTE recurrence (S7A Fig); however, significant differences were noted in asymptomatic VTE recurrence, with a marked increase from 2019 to 2020 (S7B Fig). The precise reason for the increase in asymptomatic VTE recurrence during the latter part of the study period remains

**Table 2. Cancer characteristics.**

| | Cancer group (n = 1,152) |
|---|---|
| **Cancer site** | |
| Cervical/uterine | 179 (15.5%) |
| Ovarian/peritoneal | 124 (10.8%) |
| Kidney/ureter/bladder | 112 (9.7%) |
| Central nervous system | 109 (9.5%) |
| Lung | 110 (9.5%) |
| Hematological malignancy | 107 (9.3%) |
| Breast | 67 (5.8%) |
| Colorectal | 65 (5.6%) |
| Upper gastrointestinal tract | 59 (5.1%) |
| Pancreatic | 58 (5.0%) |
| Head/neck | 54 (4.7%) |
| Liver/bile duct/gallbladder | 32 (2.8%) |
| Prostate | 30 (2.6%) |
| Other | 22 (1.9%) |
| Skin | 10 (0.9%) |
| Testicular | 10 (0.9%) |
| Unknown | 4 (0.3%) |
| **Cancer status** | |
| Distant metastasis | 390 (33.9%) |

Data are expressed as n (%).

unclear. One plausible explanation is the change in follow-up practices after DVT diagnosis, leading to more frequent use of lower limb ultrasonography at earlier time points. S8 Fig compares the number of days from the initial DVT diagnosis to the first post-diagnosis ultrasonography, which was evaluated annually. The median interval was notably shortened in 2019 and 2020, suggesting that earlier follow-up may have contributed to the increased detection of asymptomatic VTE in these years.

Cancer was significantly associated with an increased risk of VTE recurrence (HR: 2.61; 95% CI: 1.93–3.55; P < 0.001) (Table 5). Considering that mortality may occur before VTE recurrence, mortality was accounted for using the Fine–Gray subdistribution hazard model. After adjusting for confounders, cancer remained an independent predictor of VTE recurrence (adjusted sHR: 2.00; 95% CI: 1.46–2.74; P < 0.001) (Table 5). For the sensitivity analysis, we applied a Cox proportional hazards model to treat death as a censored event. Cancer remained significantly associated with VTE recurrence (adjusted HR: 2.47; 95% CI: 1.78–3.42; P < 0.001), supporting the robustness of the findings obtained from the Fine–Gray model.

Subgroup analyses using multivariable Fine–Gray models showed a consistent trend toward a higher risk of VTE recurrence in patients with cancer than in those without cancer (Fig 3). Further stratification by cancer type revealed that hematological malignancies, head/neck, cervical/uterine, ovarian/peritoneal, pancreatic, colorectal, and CNS cancers were significantly associated with a higher risk of VTE recurrence (S9 Fig). Among patients in the cancer group, those who discontinued anticoagulation therapy because of bleeding events had a significantly higher risk of recurrent VTE than those who discontinued anticoagulation therapy for other reasons (P < 0.001) (S10A Fig). Conversely, in the non-cancer group, there was no significant difference in VTE recurrence between those who discontinued treatment because of bleeding events and those who discontinued treatment for other reasons (P = 0.716) (S10B Fig).

**Table 3. Treatment strategies.**

| | Active cancer group (n = 1,152) | Non-cancer group (n = 1,129) | P-value |
|---|---|---|---|
| **Initial parenteral therapy** | | | |
| **Initial parenteral anticoagulation** | 300 (26.0%) | 248 (21.9%) | 0.023 |
| **Thrombolysis** | 2 (0.2%) | 3 (0.3%) | 0.684 |
| **Anticoagulation therapy beyond the acute phase** | 908 (78.8%) | 797 (70.6%) | <0.001 |
| **Warfarin** | 66/908 (7.3%) | 149/797 (18.7%) | <0.001 |
| **Direct oral anticoagulant** | 786/908 (86.6%) | 604/797 (75.8%) | <0.001 |
| **Edoxaban** | 678/786 (86.3%) | 482/604 (79.8%) | 0.001 |
| **Apixaban** | 69/786 (8.8%) | 70/604 (11.6%) | 0.083 |
| **Rivaroxaban** | 37/786 (4.7%) | 46/604 (7.6%) | 0.023 |
| **Dabigatran** | 2/786 (0.3%) | 6/604 (1.0%) | 0.084 |
| **Unfractionated heparin** | 56/908 (6.2%) | 43/797 (5.4%) | 0.496 |
| **Low-molecular-weight heparin** | 0/908 (0%) | 1/797 (0.1%) | 0.467 |
| **Inferior vena cava filter use** | 7 (0.6%) | 1 (<0.1%) | 0.070 |
| **Discontinuation of anticoagulation during follow-up** | 473/908 (52.1%) | 403/797 (50.6%) | 0.525 |
| **Reason for discontinuation** | | | |
| **Physician's judgment** | 329/473 (69.6%) | 345/403 (85.6%) | <0.001 |
| **Bleeding events** | 135/473 (28.5%) | 56/403 (13.9%) | <0.001 |
| **Other** | 9/473 (1.1%) | 2/403 (0.5%) | 0.176 |

Data are expressed as n (%).

**Table 4. Clinical outcomes.**

| | Overall (n = 2,281) | Active cancer group (n = 1,152) | Non-cancer group (n = 1,129) | P-value |
|---|---|---|---|---|
| **Recurrence VTE** | 192 (8.4%) | 132 (11.5%) | 60 (5.3%) | <0.001 |
| **DVT only[a]** | 169 (7.4%) | 112 (9.7%) | 57 (5.0%) | <0.001 |
| **PE with or without DVT[a]** | 23 (1.0%) | 20 (1.7%) | 3 (0.3%) | <0.001 |
| **Major bleeding** | 190 (8.3%) | 142 (12.3%) | 48 (4.3%) | <0.001 |
| **All-cause death** | 517 (22.7%) | 447 (38.8%) | 70 (6.2%) | <0.001 |
| **Causes of death** | | | | |
| **Death from cancer** | 409 (17.9%) | 401 (34.8%) | 8 (0.7%) | <0.001 |
| **Fatal PE** | 2 (<0.1%) | 1 (<0.1%) | 1 (<0.1%) | >0.999 |
| **Fatal bleeding** | 6 (0.3%) | 6 (0.5%) | 0 (0%) | 0.031 |
| **Cardiac death** | 27 (1.2%) | 13 (1.1%) | 14 (1.2%) | 0.805 |
| **Others** | 57 (2.5%) | 14 (1.2%) | 43 (3.8%) | <0.001 |
| **Unknown** | 16 (0.7%) | 12 (1.0%) | 4 (0.4%) | 0.049 |

Data are expressed as n (%).

[a]"DVT only" and "PE with or without DVT" are mutually exclusive categories.

DVT, deep vein thrombosis; PE, pulmonary embolism, VTE venous thromboembolism.

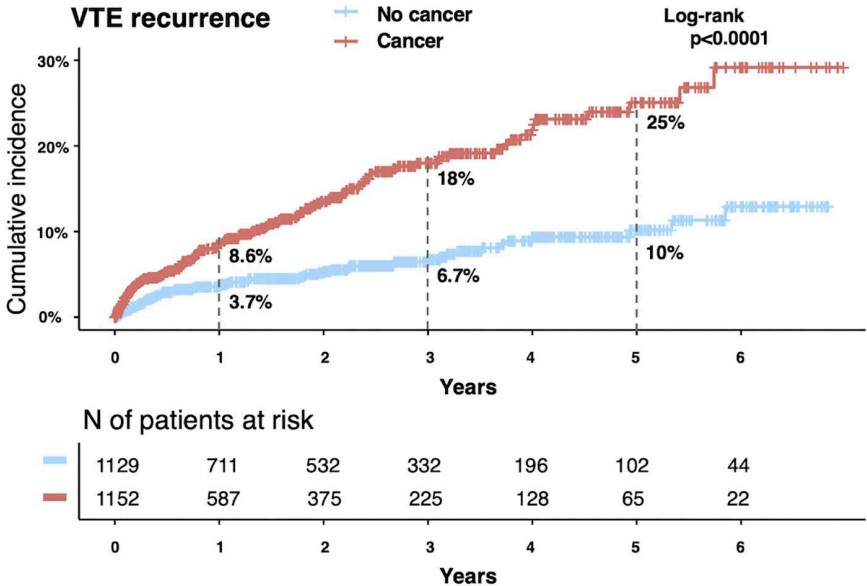

**Fig 2. Kaplan–Meier curves for VTE recurrence.** VTE, venous thromboembolism.

**Table 5. Effect of cancer on VTE recurrence.**

|  | Events, n (%) | HR (95% CI) | P-value | sHR (95% CI) | P-value | Adjusted sHR (95% CI)[a] | P-value |
|---|---|---|---|---|---|---|---|
| **Cancer** | 132 (11.5%) | 2.61 (1.93–3.55) | <0.001 | 2.00 (1.47–2.71) | <0.001 | 2.00 (1.46–2.74) | <0.001 |
| **Non-cancer** | 60 (5.3%) | 1 |  | 1 |  | 1 |  |

[a]Adjusted for age, sex, body mass index, comorbidities as a whole, baseline pulmonary embolism, hemoglobin level, anticoagulation beyond the acute phase, and discontinuation of anticoagulants.

CI, confidence interval; HR, hazard ratio; sHR, subdistribution HR; VTE, venous thromboembolism.

**Major bleeding.** During the follow-up period, 142 patients (12.3%) in the cancer group experienced major bleeding events compared with 48 patients (4.3%) in the non-cancer group (P<0.001) (Table 4). No temporal differences were observed in major bleeding events between the groups (S11 Fig). We further stratified the patients into three groups—non-cancer, cancer without distant metastasis, and cancer with distant metastasis—to explore differences in clinical outcomes. Major bleeding events occurred most frequently in patients with cancer with distant metastasis, followed by patients with cancer without metastasis and patients without cancer (S12 Fig). The gastrointestinal tract was the most common site of major bleeding in both groups, whereas the urinary tract was a more frequent site of major bleeding in the cancer group compared to the non-cancer group (Table 6). The cumulative incidence of major bleeding was significantly higher in the cancer group at 1 (15% vs. 4.8%), 3 (24% vs. 7.2%), and 5 years (30% vs. 9.6%) (log-rank P<0.001) (Fig 4). Cancer was significantly associated with a greater risk of major bleeding (HR: 3.21; 95% CI: 2.31–4.47; P<0.001) (Table 7). Even after adjusting for confounders and considering the competing risk of all-cause death, cancer remained an independent predictor of major bleeding (adjusted sHR: 2.69; 95% CI: 1.90–3.81; P<0.001), as determined by a Fine–Gray competing risk regression model (Table 7). For sensitivity analysis, we applied a Cox proportional hazards model to treat death as a censored event. Cancer remained significantly associated with major bleeding risk (adjusted HR: 2.71; 95% CI: 1.52–3.83; P<0.001), supporting the results obtained from the Fine–Gray model.

| VTE recurrence | Cancer (n = 1,152) | Non-cancer (n = 1,129) | | Adjusted sHR (95%CI) | P interaction |
|---|---|---|---|---|---|
| **Age** | | | | | |
| ≧70 years | 73/621 (11.8%) | 33/650 (5.08%) | | 2.16 (1.41–3.26) | 0.386 |
| <70 years | 59/531 (11.1%) | 27/479 (5.63%) | | 1.64 (1.01–2.66) | |
| **Sex** | | | | | |
| Male | 36/402 (8.96%) | 19/371 (5.12%) | | 1.50 (0.86–2.62) | 0.250 |
| Female | 96/750 (12.8%) | 41/758 (5.41%) | | 2.20 (1.52–3.12) | |
| **Hb** | | | | | |
| ≧10 g/dL | 104/842 (12.4%) | 45/886 (5.08%) | | 2.26 (1.56–2.54) | 0.153 |
| <10 g/dL or using EPO | 28/309 (9.06%) | 15/234 (6.41%) | | 1.40 (0.81–2.45) | |
| **CRP** | | | | | |
| ≧1.0 mg/dL | 63/573 (11.0%) | 24/521 (4.61%) | | 2.48 (1.54–4.00) | 0.200 |
| <1.0 mg/dL | 69/575 (12.0%) | 36/595 (6.05%) | | 1.66 (1.01–2.50) | |
| **Pulmonary embolism** | | | | | |
| Yes | 12/126 (9.52%) | 2/68 (2.94%) | | 2.75 (0.60–12.6) | 0.662 |
| No | 120/1026 (11.7%) | 58/1061 (5.47%) | | 1.95 (1.41–2.69) | |
| **History of VTE** | | | | | |
| Yes | 5/47 (10.6%) | 4/90 (4.44%) | | 1.56 (0.41–5.82) | 0.711 |
| No | 127/1105 (11.5%) | 56/1039 (5.39%) | | 2.01 (1.45–2.78) | |
| **Body mass index (continuous)** | | | | | |
| BMI per 1kg/m² (non-cancer) | - | 23.7 (21.0–25.6) | | 0.99 (0.91–1.07) | 0.649 |
| BMI per 1kg/m² (cancer) | 21.9 (19.6–25.4) | - | | 1.01 (0.97–1.05) | |

**Fig 3. Subgroup analyses for VTE recurrence.** The sHRs for VTE recurrence in the two groups were estimated using multivariable Fine–Gray models adjusted for age, sex, body mass index, comorbidities as a whole, baseline pulmonary embolism, hemoglobin level, anticoagulation beyond the acute phase, and discontinuation of anticoagulants. The body mass index was treated as a continuous variable. P-values for interactions were estimated to assess the heterogeneity across subgroups. CI, confidence interval; CRP, C-reactive protein; EPO, erythropoietin; Hb, hemoglobin; sHR, subdistribution hazard ratio; VTE, venous thromboembolism.

**Table 6. Sites of major bleeding.**

| | Overall (n = 190) | Active cancer group (n = 142) | Non-cancer group (n = 48) | P-value |
|---|---|---|---|---|
| **GI bleeding** | 83 (43.7%) | 63 (44.4%) | 20 (41.7%) | 0.744 |
| **Urinary bleeding** | 29 (15.3%) | 27 (19.0%) | 2 (4.2%) | 0.013 |
| **Genital bleeding** | 22 (11.6%) | 19 (13.4%) | 3 (6.2%) | 0.182 |
| **Intracranial bleeding** | 14 (7.4%) | 11 (7.7%) | 3 (6.2%) | >0.999 |
| **Airway/nasal bleeding** | 11 (5.8%) | 6 (4.2%) | 5 (10.4%) | 0.149 |
| **Skin/intramuscular bleeding** | 14 (7.4%) | 4 (2.8%) | 10 (20.8%) | <0.001 |
| **Abdominal bleeding** | 9 (4.7%) | 7 (4.9%) | 2 (4.2%) | >0.999 |
| **Other bleeding** | 8 (4.2%) | 5 (3.5%) | 3 (6.3%) | 0.419 |

Data are expressed as n (%).

GI, gastrointestinal.

Subgroup analyses using multivariable Fine–Gray models showed a consistent trend toward a higher risk of major bleeding events in patients with cancer than in patients without cancer (Fig 5). Further stratification according to cancer type revealed that pancreatic, cervical/uterine, colorectal, kidney/ureter/bladder, and ovarian/peritoneal cancers were associated with a higher risk of major bleeding (S13 Fig).

**Overall survival.** During the follow-up period, 447 patients (38.8%) in the cancer group died compared with 70 patients (6.2%) in the non-cancer group (P < 0.001). We also evaluated year-by-year trends in OS and found no temporal

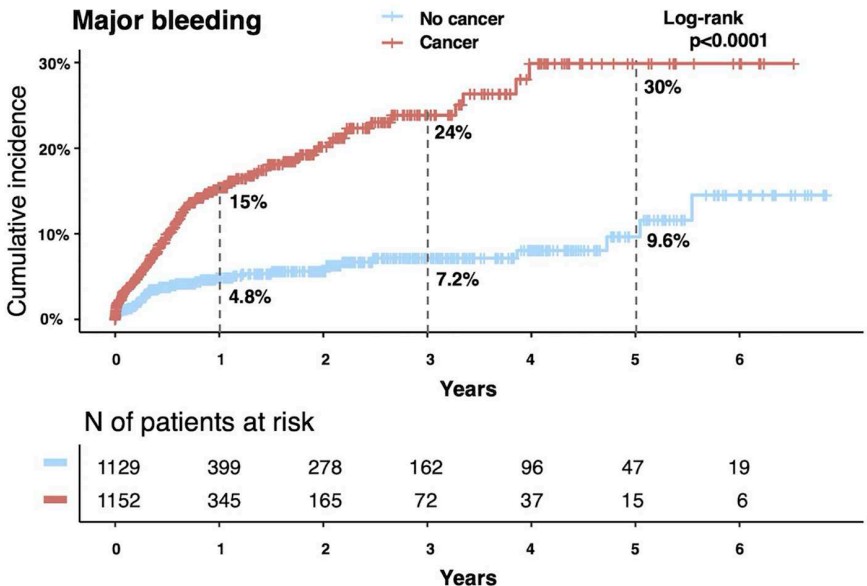

**Fig 4. Kaplan–Meier curves for major bleeding.**

**Table 7. Effect of cancer on major bleeding.**

|  | Events, n (%) | Crude HR (95% CI) | P -value | sHR (95% CI) | P-value | Adjusted sHR (95% CI) [a] | P-value |
|---|---|---|---|---|---|---|---|
| **Cancer** | 142 (12.3%) | 3.21 (2.31–4.47) | <0.001 | 2.80 (2.02–3.89) | <0.001 | 2.69 (1.90–3.81) | <0.001 |
| **Non-cancer** | 48 (4.3%) | 1 |  | 1 |  | 1 |  |

[a]Adjusted for age, sex, body mass index, comorbidities as a whole, baseline pulmonary embolism, hemoglobin level, and anticoagulation beyond the acute phase.

CI, confidence interval; HR, hazard ratio; sHR, subdistribution HR; VTE, venous thromboembolism.

differences in either group (S14 Fig). The causes of death are detailed in Table 4, with cancer progression being the most common cause in the cancer group (34.8%). The incidence of fatal PE was one in both the groups. Notably, six fatal bleeding events occurred in the active cancer group, whereas none occurred in the non-cancer group. These included intracranial hemorrhage (n = 2), gastrointestinal bleeding (n = 2), tumor-associated intraperitoneal bleeding (n = 1), and traumatic head injury (n = 1); all patients receiving anticoagulation at the time of death. Survival rates were significantly lower in the cancer group than in the non-cancer group at 1 (70% vs. 96%), 3 (54% vs. 92%), and 5 years (47% vs. 90%) (log-rank P < 0.001) (Fig 6). Interestingly, in both groups, patients who experienced major bleeding had significantly lower survival rates than those who did not. The estimated 5-year survival probability was 24% in the cancer group with major bleeding compared to 50% in those without major bleeding (P < 0.001), and 56% in the non-cancer group with major bleeding compared to 91% in those without (P < 0.001) (S15 Fig). We further investigated the impact of VTE recurrence and major bleeding on the OS of patients with and without cancer (S16 Fig). Among the cancer group, major bleeding was significantly associated with poorer survival (adjusted HR: 1.71; 95% CI: 1.33–2.19; P < 0.001), whereas VTE recurrence was not (adjusted HR: 1.06; 95% CI: 0.80–1.40; P = 0.696). In contrast, in the non-cancer group, both VTE recurrence (adjusted HR: 2.42; 95% CI: 1.16–5.05; P = 0.020), and major bleeding (adjusted HR: 5.24; 95% CI: 2.72–10.1; P = 0.043) were significantly associated with worse survival. Furthermore, we performed stratified analyses according to the cancer type. VTE recurrence was significantly associated with decreased survival in ovarian/peritoneal cancer (adjusted

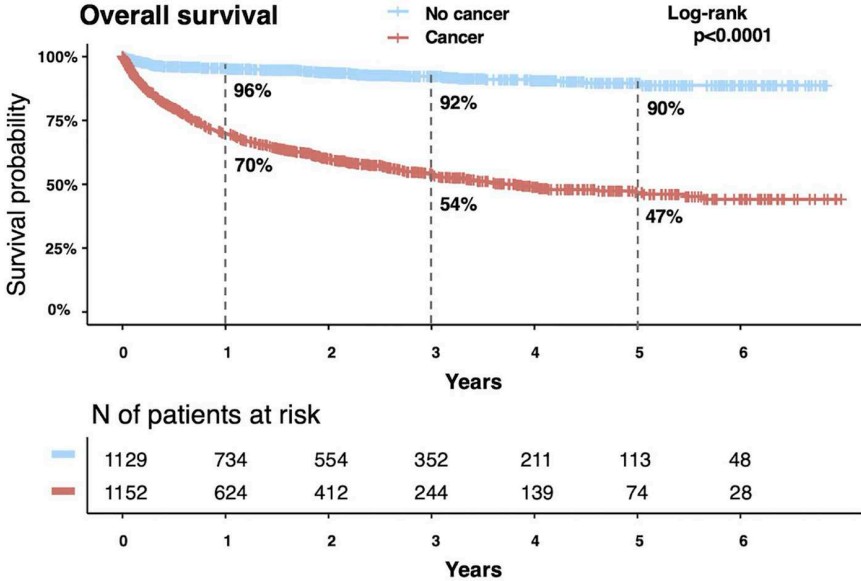

| Major bleeding events | Cancer (n = 1,152) | Non-cancer (n = 1,129) | Adjusted sHR (95%CI) | P interaction |
|---|---|---|---|---|
| **Age** | | | | |
| ≧70 years | 65/621 (10.5%) | 33/650 (5.08%) | 1.97 (1.28–3.01) | 0.035 |
| <70 years | 77/531 (14.5%) | 15/479 (3.13%) | 4.22 (2.36–7.56) | |
| **Sex** | | | | |
| Male | 35/402 (8.71%) | 18/371 (4.85%) | 1.59 (0.89–2.86) | 0.040 |
| Female | 107/750 (14.3%) | 30/758 (3.96%) | 3.32 (2.19–5.04) | |
| **Hb** | | | | |
| ≧10g /dL | 87/842 (10.3%) | 33/886 (3.72%) | 2.70 (1.71–4.26) | 0.923 |
| <10g /dL or using EPO | 55/309 (17.8%) | 15/234 (6.41%) | 2.61 (1.56–4.37) | |
| **CRP** | | | | |
| ≧1.0 mg/dL | 84/573 (14.7%) | 27/521 (5.18%) | 3.14 (2.00–4.93) | 0.325 |
| <1.0 mg/dL | 58/575 (10.1%) | 21/595 (3.53%) | 2.25 (1.35–3.76) | |
| **Pulmonary embolism** | | | | |
| Yes | 23/126 (18.3%) | 4/68 (5.89%) | 2.30 (0.79–6.69) | 0.782 |
| No | 119/1026 (11.6%) | 44/1061 (4.15%) | 2.70 (1.88–3.87) | |
| **History of VTE** | | | | |
| Yes | 5/47 (10.6%) | 1/90 (1.11%) | 4.79 (0.55–41.6) | 0.588 |
| No | 137/1105 (12.4%) | 47/1039 (4.52%) | 2.62 (1.84–3.72) | |
| **Body mass index (continuous)** | | | | |
| BMI per 1kg/m² (non-cancer) | - | 22.3 (19.7–24.9) | 0.98 (0.91–1.07) | 0.606 |
| BMI per 1kg/m² (cancer) | 22.1 (18.8–25.6) | - | 1.01 (0.97–1.05) | |

**Fig 5. Subgroup analyses for major bleeding.** The sHRs for major bleeding in the two groups were derived using multivariable Fine–Gray models adjusted for age, sex, body mass index, comorbidities as a whole, baseline pulmonary embolism, hemoglobin level, and anticoagulation beyond the acute phase. Body mass index was treated as a continuous variable. P-values for interactions were estimated to assess heterogeneity across subgroups. CI, confidence interval; CRP, C-reactive protein; EPO, erythropoietin; Hb, hemoglobin; sHR, subdistribution hazard ratio; VTE, venous thromboembolism.

**Fig 6. Kaplan–Meier curves for overall survival.**

HR: 2.91; 95% CI: 1.52–5.59; P<0.001), and major bleeding was strongly associated with poorer survival in kidney/ureter/bladder cancer (adjusted HR: 2.77; 95% CI: 1.07–7.19; P=0.037), and upper gastrointestinal tract cancer (adjusted HR: 3.88; 95% CI: 1.08–13.9; P=0.039) (S17 Fig).

## Discussion

The main findings of this study were as follows: (1) Patients with cancer-associated VTE had a significantly higher risk of VTE recurrence and major bleeding than those without cancer, even after adjusting for confounders and accounting for the competing risk of all-cause mortality; (2) OS was worse in VTE patients with cancer compared to those without cancer; (3) The discontinuation rate of anticoagulation therapy during follow-up was similar between the cancer and non-cancer groups; however, the reasons for discontinuation differed, with a higher proportion of discontinuations because of bleeding in the cancer group; (4) Among patients in the cancer group, those who discontinued anticoagulation because of bleeding had a significantly higher risk of recurrent VTE compared to those who discontinued for other reasons; (5) VTE patients with or without cancer who experienced major bleeding had poorer OS.

In this study, we demonstrated that even in the DOAC era, patients with cancer remain at a higher risk for VTE recurrence than patients without cancer, a finding consistent with previous reports from the warfarin era [6,15–21]. However, real-world data on CAT use in the context of DOAC use remain limited. A recent large-scale registry study by Chatani et al. reported that the unadjusted risk of VTE recurrence is slightly higher in patients with cancer. However, after adjusting for confounders and accounting for competing mortality risks, the long-term recurrence risk was similar between patients with and without cancer [7]. This contradicts our findings, and the discrepancy may stem from the differences in the patient selection criteria between the two studies. Chatani et al.'s study included only patients with acute symptomatic VTE, whereas our study enrolled patients diagnosed with DVT using ultrasonography, including asymptomatic patients [7]. Consequently, our cohort included fewer cases of PE and proximal DVT than other studies. Moreover, ultrasound in patients without cancer was likely performed based on symptoms and not routine screening, possibly leading to more clinically relevant VTE. Furthermore, in the study by Chatani et al. [7], patients were classified into cancer and non-cancer groups solely based on the presence of active cancer at the time of VTE diagnosis. Therefore, the non-cancer group likely included patients without any history of malignancy and those with a prior history of non-active cancer. In contrast, our non-cancer group was defined strictly as patients without a known history of cancer. This difference in group definitions may have influenced the baseline characteristics and clinical outcomes, and should be considered when comparing the two studies.

Furthermore, in our registry, 75% of patients continued anticoagulation therapy beyond the acute phase, whereas in Chatani et al.'s study, this proportion was higher (92%). Nevertheless, our subgroup analysis restricted to patients who received anticoagulation therapy still showed a higher recurrence rate of VTE in patients with cancer. Although our study focused on the DOAC era, 9.4% of the patients were still treated with warfarin. Notably, this proportion is consistent with real-world data from the COMMAND VTE Registry-2, which reported that 10.9% of patients with cancer-associated VTE received warfarin during the same period (2015–2020) [7]. Therefore, we believe that our treatment distribution accurately reflects the contemporary prescription patterns in Japan. However, the inclusion of a minority of warfarin users may limit the generalizability of our findings to a purely DOAC–treated population. In this context, our results support the view that patients with cancer remain at an increased risk of VTE recurrence, regardless of the anticoagulant type. These findings highlight the need for continued vigilance in managing VTE recurrence in patients with cancer, even in the era of DOAC therapy.

This study also demonstrated a higher risk of major bleeding in patients with cancer than in those without cancer, which is consistent with previous findings from both the warfarin and DOAC eras [6,7,15,17,20,22]. Although cancer-related deaths accounted for the majority of mortalities in the active cancer group, six patients experienced fatal bleeding during anticoagulation therapy. These included intracranial hemorrhage, gastrointestinal bleeding, and tumor-associated

vascular rupture. These findings highlight the complex clinical dilemma of balancing thrombosis and bleeding risks in patients with advanced malignancy and underscore the importance of individualized anticoagulation strategies in this vulnerable population. Current guidelines recommend the use of low-molecular-weight heparin (LMWH) or DOACs to manage CAT [3–5]. Previous studies comparing DOACs with LMWH have shown that DOACs are not inferior to dalteparin in preventing VTE recurrence in patients with cancer, without a significantly increased risk of major bleeding [12,23–25]. However, caution is advised in patients with gastrointestinal and urological cancers because of the increased bleeding risk associated with DOAC therapy. In this study, patients with gynecological, gastrointestinal, and urological cancers had a particularly high-risk of bleeding, with a significantly higher incidence of urinary tract bleeding in the cancer group than in the non-cancer group. In cases of malignant tumors in the pelvic region, radiotherapy can lead to mucosal damage and the development of hemorrhagic cystitis (HC). Radiation-induced HC primarily results from bladder mucosal injury, leading to edema and inflammation. Although most cases resolve spontaneously with a favorable prognosis, severe cases may require treatment interruption [26]. The incidence of post-radiotherapy HC in patients with prostate cancer was 11.1%, with 52% of patients requiring blood transfusions. Fatal outcomes have also been reported [27]. Similarly, pelvic irradiation for gynecological cancers poses a significant risk of HC, with severe cases leading to substantial therapeutic challenges, prolonged hospitalization, or mortality [28]. Given that pelvic radiotherapy is a recognized risk factor for HC, clinicians must carefully evaluate bleeding risk before initiating anticoagulant therapy. To minimize the risk of severe hemorrhagic complications, patients with cancer receiving anticoagulants and scheduled for pelvic irradiation should undergo a thorough bleeding risk assessment, including the use of validated bleeding risk scores [29]. However, in our retrospective study, the application of validated bleeding risk scores, such as the HAS-BLED, was not feasible owing to missing data.

In this study, the discontinuation rate of anticoagulation therapy at one year was high, although no significant difference was observed between the cancer and non-cancer groups (52% vs. 53%, P = 0.444). However, these findings should be interpreted with caution. In the non-cancer group, discontinuation at 3–6 months may be appropriate and consistent with the clinical guidelines for provoked VTE. In contrast, CAT often requires extended or indefinite anticoagulation therapy because of ongoing risk factors. Therefore, comparable discontinuation rates may reflect differences in treatment strategies, rather than true similarities in early termination. This limitation should be considered when interpreting these findings. In addition, the reasons for discontinuation varied significantly, with a higher proportion of discontinuations owing to bleeding in the cancer group than in the non-cancer group (29% vs. 14%, P < 0.001), which is consistent with previous findings from both the warfarin and DOAC eras [7,15]. This finding underscores the need for improved bleeding risk stratification in patients with cancer receiving anticoagulation therapy. Recent studies have shown that pharmacokinetic interactions between DOACs and anticancer agents exist, but most are mild, and treatment modifications are generally recommended only in clinically significant cases [30]. In contrast, certain anticancer drugs have been associated with an increased risk of bleeding when co-administered with DOACs [31], highlighting the importance of individualized assessment that considers factors such as renal dysfunction, weight loss, potential drug–drug interactions with anticancer agents, and appropriate dose adjustment in patients with cancer. Notably, patients with cancer who discontinued anticoagulation therapy owing to bleeding had a significantly higher risk of recurrent VTE than those who discontinued anticoagulation therapy for other reasons; this pattern was not observed in patients without cancer. These findings suggest the presence of shared risk factors for VTE recurrence and bleeding events in patients with cancer. The risk of bleeding may increase as cancer progresses. Previous studies examining risk factors for major bleeding in patients with VTE have identified terminal cancer and a life expectancy of ≤ 6 months as independent predictors of major bleeding [6,32,33]. In contrast, patients with terminal cancer also have an elevated risk of thrombosis because of disease progression, reduced mobility, dehydration, and infection [34]. Therefore, patients who discontinue anticoagulation therapy because of bleeding should be closely monitored to recognize their heightened vulnerability to VTE recurrence. In such cases, alternative management strategies, such as resuming anticoagulation with a reduced DOAC dose or considering non-pharmacological interventions, such as

inferior vena cava filters, may be required. Clinicians should carefully consider each patient's underlying condition, renal function, personal preferences, and lifestyle when determining the most appropriate treatment approach [35].

Our study demonstrated that patients who discontinued anticoagulants because of bleeding experienced frequent VTE recurrence, and that those who had major bleeding events had poorer OS. Additionally, a recent RCT (CANVAS trial), which aimed to address the variability in drug selection and included flexible enrolment criteria, reported a high incidence of major bleeding, with rates of 5.2% in the DOAC group and 5.6% in the LMWH group [36]. Current guidelines recommend a minimum of 6 months of anticoagulation for CAT, with extended therapy advised in the presence of persistent risk factors, such as active cancer, metastasis, or ongoing chemotherapy [3]. As outlined in the 2023 ASCO guideline, LMWH or direct oral factor Xa inhibitors are recommended for the treatment of CAT [4]. However, careful assessment of the bleeding risk is essential in patients with gastrointestinal or genitourinary malignancies. Clinicians are increasingly challenged to maintain effective anticoagulation therapy while minimizing bleeding complications in high-risk populations.

Therefore, the development of anticoagulants with a lower bleeding risk and improved adherence is essential. Factor XI (FXI) has emerged as a promising therapeutic target for CAT because it plays a crucial role in clot formation and has a minimal impact on overall coagulation and hemostasis [37,38]. Currently, FXI inhibitors are being investigated in various clinical settings, including total knee arthroplasty, atrial fibrillation, non-cardioembolic stroke, end-stage renal disease, myocardial infarction, coronavirus disease 2019, and cancer-associated VTE [39]. For the treatment of cancer-associated VTE, two ongoing phase III multicenter randomized controlled trials (RCTs) have evaluated abelacimab, a fully human monoclonal antibody: the ASTER trial (comparing abelacimab to apixaban; NCT05171049) and the MAGNOLIA trial (comparing abelacimab to dalteparin; NCT05171075). The results of these trials may help address the unmet needs in the current management of CAT.

## Study strength and limitation

This study had several notable strengths. First, it leveraged a large real-world cohort from a tertiary academic center in Japan to provide a contemporary data on cancer-associated VTE during the DOAC era. Second, unlike previous studies that primarily focused on symptomatic VTE, our cohort included both symptomatic and asymptomatic patients, particularly those with isolated distal DVT, which is frequently encountered in oncology settings. Third, we used competing risk models (Fine–Gray) to adjust for the high mortality in patients with cancer, improving the accuracy of VTE and bleeding risk estimates. Fourth, we analyzed the reasons for anticoagulant discontinuation in detail and identified bleeding-related discontinuation as a key predictor of VTE recurrence. Finally, cancer-type–specific risks for both VTE recurrence and major bleeding were explored, offering insights into personalized anticoagulation strategies in patients with cancer.

This study had some limitations. First, as a single-center study, the prevalence of the disease may have been influenced by regional and institutional factors, potentially introducing a bias. Additionally, the retrospective nature of our study and the inclusion of both symptomatic and asymptomatic DVT cases, some of which were detected incidentally during routine ultrasonography, may have limited the generalizability of our findings to populations with acute symptomatic VTE. Furthermore, owing to the retrospective design, follow-up practices after the initial DVT diagnosis were at the discretion of individual physicians, and no standardized protocol was employed, which may have influenced the outcomes of this study. Second, the implementation and duration of anticoagulation therapy were determined at the discretion of the treating physicians, indicating that individual clinical judgment and experience may have affected the outcomes. Third, although this study was conducted in the post-DOAC era, 9.4% of the overall cohort received warfarin, which may have affected the generalizability of the findings. Fourth, as the study included patients diagnosed with DVT using ultrasonography, it included asymptomatic cases and a relatively high proportion of patients with isolated distal DVT. Fifth, although we obtained referral letters and discharge summaries from patients who experienced major clinical events outside our institution, we did not conduct telephone- or letter-based follow-ups. This may have resulted in the incomplete ascertainment of clinical outcomes. Finally, although we collected information on cancer type and the presence or absence of distant

metastasis at baseline, we did not have access to detailed information regarding cancer stage or treatment. Consequently, we were unable to assess the influence of these factors on clinical outcomes.

## Conclusion

A real-world registry study in the DOAC era demonstrated a higher risk of VTE recurrence and major bleeding in patients with active cancer than in those without cancer, highlighting that these remain significant clinical challenges. Targeted efforts are necessary to mitigate the risks of VTE recurrence and bleeding, particularly in patients with cancer. These findings underscore the need for novel anticoagulant agents with improved safety profiles for CAT management.

## Supporting information

**S1 Fig. Annual trends in anticoagulation therapy beyond the acute phase.** Cancer (A) and non-cancer (B) groups. DOAC, direct oral anticoagulant; LMWH, low-molecularweight heparin; UFH, unfractionated heparin.
(TIF)

**S2 Fig. Kaplan–Meier curves for the discontinuation of anticoagulation therapy.**
(TIF)

**S3 Fig. Kaplan–Meier curves for the discontinuation of anticoagulation therapy by year.** Cancer (A) and non-cancer (B) groups.
(TIF)

**S4 Fig. Kaplan–Meier curves for VTE recurrence among patients receiving anticoagulation therapy.** VTE, venous thromboembolism.
(TIF)

**S5 Fig. Kaplan–Meier curves for VTE recurrence in non-cancer, cancer with distant metastasis, and cancer without distant metastasis.** VTE, venous thromboembolism.
(TIF)

**S6 Fig. Kaplan–Meier curves for VTE recurrence by year.** Cancer (A) and non-cancer (B) groups. VTE, venous thromboembolism.
(TIF)

**S7 Fig. Kaplan–Meier curves for VTE recurrence by year in the cancer group.** Symptomatic (A) and asymptomatic (B) VTE. VTE, venous thromboembolism.
(TIF)

**S8 Fig. Timing of follow-up venous ultrasound after the initial DVT diagnosis.** DVT, deep vein thrombosis.
(TIF)

**S9 Fig. Multivariable Fine–Gray model for VTE recurrence risk by cancer type.** The Fine–Gray models were adjusted for age, sex, body mass index, comorbidities as a whole, baseline pulmonary embolism, hemoglobin level, anticoagulation beyond the acute phase, and discontinuation of anticoagulants. CI, confidence interval; CNS, central nervous system; sHR, subdistribution hazard ratio; UGI, upper gastrointestinal; VTE, venous thromboembolism.
(TIF)

**S10 Fig. Kaplan–Meier event curves for VTE recurrence based on the reason for anticoagulation discontinuation.** Cancer (A) and non-cancer (B) groups. VTE, venous thromboembolism.
(TIF)

**S11 Fig. Kaplan–Meier curves for major bleeding by year.** Cancer (A) and non-cancer (B) groups.
(TIF)

**S12 Fig. Kaplan–Meier curves for major bleeding in non-cancer, cancer with distant metastasis, and cancer without distant metastasis.**
(TIF)

**S13 Fig. Multivariable Fine–Gray model for major bleeding risk by cancer type.** The Cox models were adjusted for age, sex, body mass index, comorbidities as a whole, baseline pulmonary embolism, hemoglobin level, and anticoagulation beyond the acute phase. CI, confidence interval; CNS, central nervous system; HR, hazard ratio; UGI, upper gastrointestinal.
(TIF)

**S14 Fig. Kaplan–Meier curves for overall survival by year.** Cancer (A) and non-cancer (B) groups.
(TIF)

**S15 Fig. Kaplan–Meier curves for overall survival based on major bleeding events.** Cancer (A) and non-cancer (B) groups.
(TIF)

**S16 Fig. Impact of VTE recurrence and major bleeding on overall survival in patients with and without cancer.** The Cox models were adjusted for age, sex, body mass index, comorbidities as a whole, baseline pulmonary embolism, hemoglobin level, anticoagulation beyond the acute phase, and discontinuation of anticoagulants, with either VTE recurrence or major bleeding included as an additional covariate in each model. CI, confidence interval; HR, hazard ratio; VTE, venous thromboembolism.
(TIF)

**S17 Fig. Impact of VTE recurrence and major bleeding on overall survival in patients with cancer, stratified by cancer type.** The Cox models were adjusted for age, sex, body mass index, comorbidities as a whole, baseline pulmonary embolism, hemoglobin level, anticoagulation beyond the acute phase, and discontinuation of anticoagulants, with either VTE recurrence or major bleeding included as an additional covariate in each model. CI, confidence interval; CNS, central nervous system; HR, hazard ratio; UGI, upper gastrointestinal; VTE, venous thromboembolism.
(TIF)

## Acknowledgments

The authors would like to thank Editage (www.editage.jp) for the English language editing.

## Author contributions

**Conceptualization:** Kazuko Tajiri.

**Data curation:** Kazuko Tajiri, Hiroyuki Naito, Momoko Murata.

**Formal analysis:** Yuki Ishizuka, Kazuko Tajiri, Masayuki Hattori.

**Funding acquisition:** Kazuko Tajiri.

**Investigation:** Yuki Ishizuka, Kazuko Tajiri, Hiroyuki Naito, Momoko Murata, Masayuki Hattori.

**Methodology:** Kazuko Tajiri, Hiroyuki Naito, Masayuki Hattori.

**Project administration:** Kazuko Tajiri.

**Supervision:** Tomoko Machino-Ohtsuka, Tomoko Ishizu.

**Visualization:** Yuki Ishizuka.

**Writing – original draft:** Yuki Ishizuka.

**Writing – review & editing:** Kazuko Tajiri, Hiroyuki Naito, Momoko Murata, Masayuki Hattori, Tomoko Machino-Ohtsuka, Tomoko Ishizu.

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
