## [Decision Letter · Decision Letter 0]

Dear Dr. Tajiri,

Thank you for submitting your manuscript to PLOS ONE. After careful consideration, we feel that it has merit but does not fully meet PLOS ONE’s publication criteria as it currently stands. Therefore, we invite you to submit a revised version of the manuscript that addresses the points raised during the review process.

**ACADEMIC EDITOR:**

Thank you for submitting your work to PLOS One. This study addresses an important clinical issue in cancer-associated thrombosis (CAT) management by providing real-world data on the risks of VTE recurrence and bleeding in the DOAC era. While interesting, based on review of your work and taking into account the feedback from reviewers, a major revision is required. I invite you to provide a point-by-point rebuttal addressing the points raised. Some additional feedback have been outlined below:

Major concerns:

The temporal trends in anticoagulation use or outcomes over the study period (2015–2020) are not examined. For example, did DOAC adoption increase over time, and were outcomes better in more recent years?

The authors do not analyse the role of cancer stage or treatment type (e.g., chemotherapy vs. radiation) in VTE recurrence or bleeding risks. These factors likely contribute to the observed differences in outcomes.

The subgroup analyses of cancer type lack adjustments for multiple comparisons. This undermines the reliability of some findings, such as the higher recurrence risk in CNS cancer patients.

The survival analyses focus primarily on differences between cancer and non-cancer groups but do not explore predictors of survival within the cancer group. For example, how does the presence of major bleeding or VTE recurrence impact survival in different cancer subtypes?

Besides, some specific comments are outlined below for authors to consider:

1. In the abstract, details on subgroup findings or the implications of anticoagulation discontinuation, which are important outcomes, are lacking. Please include.

2. The introduction does not clearly differentiate how the study builds upon existing literature, such as the COMMAND VTE Registry studies. The rationale for focusing on cancer patients with DVT diagnosed via ultrasound is not sufficiently justified. A discussion of why asymptomatic cases are important to study would provide more clarity.

3. The study excludes patients with DVT diagnosed before 2015 to focus on the DOAC era but does not clarify whether warfarin use in 9.4% of the cohort is representative of contemporary practice. This could bias the findings. Please explain.

4. While the Fine-Gray subdistribution hazard model accounts for competing risks, the authors does not report sensitivity analyses to test the robustness of its findings. This is critical given the high mortality rate in the cancer group.

5. The study does not mention whether validated bleeding risk scores were used to adjust for bleeding risk in the cancer cohort. This could affect the accuracy of the hazard ratios.

6. Subgroups based on cancer type are analyzed but not adjusted for multiple comparisons. This raises the risk of type I error. This should be addressed.

7. Table 1: While the baseline characteristics are comprehensive, the authors do not report p-values for all comparisons, such as DVT location. Any non-significant findings should still be reported.

8. Table 4: The all-cause mortality data is significant, but the breakdown of causes of death is underexplored in the discussion. For example, the higher rate of cancer-related deaths in the cancer group is expected, but the significance of fatal bleeding events in cancer patients warrants further elaboration.

9. Figure 3 and 5: Subgroup analyses are presented without adjustments for multiple comparisons. The authors should clarify this limitation in the results section.

10. The similar rates of anticoagulation discontinuation between groups are reported without sufficient discussion of the clinical implications. For example, the higher rate of discontinuation due to bleeding in the cancer group suggests a need for better bleeding risk stratification.

11. The discussion does not adequately address the limitations of the study design, particularly the retrospective nature, single-center setting, and the inclusion of asymptomatic DVT cases. Besides, the potential impact of warfarin use in a minority of patients is not discussed in detail. How might this affect the generalizability of the results to a purely DOAC-treated population?

The discussion could expand on the implications of VTE recurrence in patients who discontinued anticoagulation due to bleeding. For instance, should alternative anticoagulants (e.g., low-dose DOACs) or non-pharmacologic strategies be considered in these patients?

Authors should also consider expanding on the VTE/DVT management (see Sun et al PMID: 37175686)

We look forward to receiving your revised manuscript.

Kind regards,

Sonu Bhaskar, MD PhD

Academic Editor

PLOS ONE

**Journal Requirements:**

1. When submitting your revision, we need you to address these additional requirements. Please ensure that your manuscript meets PLOS ONE's style requirements, including those for file naming. The PLOS ONE style templates can be found at https://journals.plos.org/plosone/s/file?id=wjVg/PLOSOne_formatting_sample_main_body.pdf and https://journals.plos.org/plosone/s/file?id=ba62/PLOSOne_formatting_sample_title_authors_affiliations.pdf 2. Thank you for stating the following in the Competing Interests section: I have read the journal's policy and the authors of this manuscript have the following competing interests: KT has received honoraria from Bristol Myers Squibb, Pfizer, Bayer, and Daiichi Sankyo. The remaining authors declare that the research was conducted in the absence of any commercial or financial relationships that could be construed as a potential conflict of interest. We note that one or more of the authors are employed by a commercial company.  a. Please provide an amended Funding Statement declaring this commercial affiliation, as well as a statement regarding the Role of Funders in your study. If the funding organization did not play a role in the study design, data collection and analysis, decision to publish, or preparation of the manuscript and only provided financial support in the form of authors' salaries and/or research materials, please review your statements relating to the author contributions, and ensure you have specifically and accurately indicated the role(s) that these authors had in your study. You can update author roles in the Author Contributions section of the online submission form. Please also include the following statement within your amended Funding Statement. “The funder provided support in the form of salaries for authors [insert relevant initials], but did not have any additional role in the study design, data collection and analysis, decision to publish, or preparation of the manuscript. The specific roles of these authors are articulated in the ‘author contributions’ section.”If your commercial affiliation did play a role in your study, please state and explain this role within your updated Funding Statement.  b. Please also provide an updated Competing Interests Statement declaring this commercial affiliation along with any other relevant declarations relating to employment, consultancy, patents, products in development, or marketed products, etc.   Within your Competing Interests Statement, please confirm that this commercial affiliation does not alter your adherence to all PLOS ONE policies on sharing data and materials by including the following statement: "This does not alter our adherence to  PLOS ONE policies on sharing data and materials.” (as detailed online in our guide for authors http://journals.plos.org/plosone/s/competing-interests) . If this adherence statement is not accurate and  there are restrictions on sharing of data and/or materials, please state these. Please note that we cannot proceed with consideration of your article until this information has been declared. Please include both an updated Funding Statement and Competing Interests Statement in your cover letter. We will change the online submission form on your behalf. 3. In the online submission form, you indicated that The data underlying the results presented in the study are available from the corresponding author on reasonable request. All PLOS journals now require all data underlying the findings described in their manuscript to be freely available to other researchers, either a. In a public repository, b. Within the manuscript itself, or c. Uploaded as supplementary information.This policy applies to all data except where public deposition would breach compliance with the protocol approved by your research ethics board. If your data cannot be made publicly available for ethical or legal reasons (e.g., public availability would compromise patient privacy), please explain your reasons on resubmission and your exemption request will be escalated for approval.

Reviewers' comments:

Reviewer's Responses to Questions

**Comments to the Author**

1. Is the manuscript technically sound, and do the data support the conclusions?

Reviewer #1: Yes

Reviewer #2: Yes

2. Has the statistical analysis been performed appropriately and rigorously?

Reviewer #1: Yes

Reviewer #2: Yes

3. Have the authors made all data underlying the findings in their manuscript fully available?

Reviewer #1: Yes

Reviewer #2: Yes

4. Is the manuscript presented in an intelligible fashion and written in standard English?

Reviewer #1: Yes

Reviewer #2: Yes

**Reviewer #1:**  Thank you very much for providing the opportunity to review the manuscript. It is written in a straight forward manner, with a clear research question, a suitable method and statistics and a clear result. There is littele to discuss or to amend. Results seems reasonable. It would have been interesting to compare the 4 DOAC among each other regarding thrombosis and bleeding. Manybe a follow-up project?

There are very few typos, which can be corrected upon processing.

**Reviewer #2: ** Introduction:

Problem statement: Not clear. Please spell it out in a stand-alone paragraph.

Study objectives: Not clear. Please spell it out in a stand-alone paragraph. Please spell out the primary and secondary objectives.

Methods:

Line 62 – 63: The registry is a single-center, retrospective cohort study.

Naturally when we do a registry, we either enter data on the spot or later, some may have a follow-up. Therefore, please explain why this registry is retrospective, or otherwise.

Line 74: Among the 17,188 patients enrolled in the TULIPE registry.

These are adult patients only or include children. If adults only, please define the cut-off age.

Line 76 – 78: all the exclusion criteria.

Do all samples have complete data that are required for the study? If yes, then ignore my comment. If no, please add it as an exclusion criteria.

Please specify in methodology that the study data was extracted from the TULIPE registry and divided into different sections: Demographic, clinical, investigations, treatment, outcomes… This corresponds to the results reported latter.

The variables for the study were not stated in the methodology. Please spell it out.

Example: Patient demographic, clinical, laboratory, and treatment data was collected from the registry.

The demographic data include age, gender, BMI…The clinical data include comorbidity, types of DVT, presence of PE… For investigations, it will be better to highlight parameters associated with VTE are selected (this explains why CRP included).

Line 80 – 83: The cancer group was defined as patients with a cancer diagnosis within 6 months prior to the ultrasound; recurrent, regionally advanced, or metastatic cancer; cancer treatment within the previous 6 months; or hematologic cancer not in complete remission (as previously described) [7]

The above definition refers to ‘active cancer’ by Raskob et al. However, they also include cancer within 2 years ‘Patients had to have cancer other than basal-cell or squamous-cell skin cancer that was active or had been diagnosed within the previous 2 years and was objectively confirmed’. Therefore, the cancer group in this study is better renamed as active cancer group.

Line 90 – 91: VTE was defined as new-onset lower extremity venous thrombosis, pulmonary embolism (PE), or Trousseau syndrome, confirmed by imaging.

Please cite the articles to support this definition.

Overall survival was not defined. Let's say patients die at home or transfer care to another centre and die, does this data capture in your study?

Table 3: Anticoagulation therapy beyond the acute phase.

Please define the acute phase in methodology.

Results:

It will be more organized to divide the results into 2 main sections:

1. Baseline: Demographic, clinical, and treatments.

2. Outcomes: VTE recurrence, Bleed, Mortality

So overall results flow is better.

Line 118 – 126: Please specify the value (%) followed by p for the variables with significant differences. It was not stated for this part but mentioned for the remaining parts of the result. Please standardise it.

Line 164 – 165, and Line 202 – 204: The adjusted ratio for VTE recurrence and major bleed was based on age, gender, and BMI.

By right, the adjusted ratio should be based on covariates with p < 0.05 and deemed significant in other studies. Therefore, in my opinion, it should be:

Age, BMI (continuous variables, not BMI as the number is too small), comorbid as a whole, baseline PE, HB level, anticoagulant beyond acute phase, and discontinuation of anticoagulant (even though p not significant, I believe must be related to recurrence of VTE and major bleed in other studies)

Please do the same correction for OS, it was not mentioned in the result.

Discussion:

Line 261 – 264: Chatani et al.'s study included only patients with acute symptomatic VTE, whereas our study enrolled patients diagnosed with DVT via ultrasonography, including asymptomatic cases and those with an unknown onset time. Consequently, our cohort had fewer patients with PE or proximal DVT compared to Chatani et al.’s study.

By right, Chatani et al cancer + acute vs acute compared to current study cancer + scan vs scan (those non-cancer groups unlikely have a routine scan, they scan most likely something going on). The outcome should be worse in Chatani. I would rather argue that Chatani et al include active cancer + cancer 6-24 months vs the current study that includes active cancer, leading to dilution of differences in the former. Please read Chatani et al again and see if justifiable. If so, please write the discussion again.

Line 292 – 294: In this study, the discontinuation rate of anticoagulation therapy at one year was high,

although no significant difference was observed between the cancer and non-cancer groups (52%

vs. 53%, P = 0.444).

I would rather argue that discontinuation for non-cancer is falsely high and leads to no different in result. In the non-cancer group, some patients may have discontinuation at 3 to 6 months due to standard practice, while cancer is often life-long. If the number of patients in non-cancer only takes into those who discontinued prematurely before the appropriate period, the exact number may be smaller.

Line 315 – 325: The development of anticoagulants with a lower bleeding risk and improved

316 adherence is essential. Factor XI (FXI) has emerged as a promising therapeutic target for CAT…

This is the clinical implication of this study, please put in a stand alone paragraph.

Study strength: Not mentioned. Please highlight.

Conclusion:

Maybe mentioned a newer agent to address these issues is required?

Tables:

Table 1:

A BMI of more than 35 can remove as the number too small.

Comorbidities as overall, the p should be presented.

For diseases that cannot recover, like chronic lung, heart disease… Please remove the history of, as the disease is still present. Maybe for VTE only is suitable.

Presentation: DVT should be put above, and PE below.

Table 3:

Ventilator support to be removed. It is irrelevant.

Table 4:

DVT should be above PE. Why DVT only, PE with or without DVT? It is rather confusing.

Could it be divided into DVT and PE? Or DVT, PE, DVT + PE?

Tables 5 and 7: Suggest to correct, including relevant covariates as mentioned in the comment above.

Figures:

Figure 3 and 5: Why these parameters are selected?

TWC and creatinine clearance were not even mentioned in Table 1. Some don’t have between-group significance like gender, or platelet. The cut-off point of these parameters is also not justified.

I would suggest using the parameters selected for the adjusted ratio as mentioned above, such as:

Age, BMI (continuous variables, not BMI as the number is too small), comorbid as a whole, baseline PE, HB level, anticoagulant beyond acute phase, and discontinuation of anticoagulant (even though p not significant, I believe must be related to recurrence of VTE and major bleed in other studies)\

**Do you want your identity to be public for this peer review?** For information about this choice, including consent withdrawal, please see our Privacy Policy

Reviewer #1: **Yes: ** Olaf Rose

Reviewer #2: **Yes: ** Diana-Leh-Ching Ng

---

## [Author Response · Author response to Decision Letter 1]

15 Jun 2025

Dear Editors and Reviewers

We sincerely appreciate the time and effort spent reviewing and providing valuable comments on our manuscript. We are grateful to the reviewers for their favorable comments and valuable suggestions, which have helped us significantly improve our manuscript. As indicated in the following responses, we have carefully considered all the comments and suggestions in the revised version of our paper. These changes are highlighted in yellow in the revised manuscript. We have incorporated all the suggestions and believe that they have significantly improved the scientific content of the manuscript. Thank you once again.

Response to the Academic Editor

Major comment #1:

The temporal trends in anticoagulation use or outcomes over the study period (2015–2020) are not examined. For example, did DOAC adoption increase over time, and were outcomes better in more recent years?

Reply:

As suggested, we have analyzed the temporal trends in anticoagulant use during the study period. DOAC use increased from 2015 to 2020 in both the cancer and non-cancer groups (S1A Fig and S1B Fig), whereas the use of warfarin and other anticoagulants declined during the same period. We have added this information to the “Treatment strategies” section, as follows:

The trends in anticoagulant use during the study period are presented in S1 Fig. In both the cancer and non-cancer groups, the percentage of DOAC prescriptions increased over time, whereas the percentage of prescriptions for warfarin and other anticoagulants decreased. (page 13, lines 203-205)

We also performed Kaplan–Meier analyses to evaluate the temporal trends in the discontinuation of anticoagulation therapy, VTE recurrence, major bleeding, and overall survival. There were no significant differences in the rate of anticoagulation discontinuation across calendar years in either group (S3 Fig). We have added this information to the “Treatment strategies” section of the revised manuscript as follows:

There was no significant difference in the annual rate of anticoagulation therapy discontinuation during the study period. (S3 Fig). (page 14, lines 209-210)

We conducted Kaplan–Meier analyses of VTE recurrence by year (S6 Fig). In the cancer group, the cumulative incidence of VTE recurrence varied significantly by year, with notably higher recurrence rates observed in the latter part of the study period (2019–2020). No significant differences were observed in the VTE recurrence by year in the non-cancer group. To explore the factors contributing to these annual differences within the cancer group, we conducted Kaplan–Meier analyses, stratifying cases into symptomatic and asymptomatic VTE recurrences (S7 Fig). There was no significant difference in symptomatic VTE recurrence by year (S7A Fig); however, there were significant differences in asymptomatic VTE recurrence, with a marked increase from 2019 to 2020 (S7B Fig).

The precise reason for the increased recurrence of asymptomatic VTE during the latter part of the study period remains unclear. One plausible explanation is a change in follow-up practices after DVT diagnosis, leading to the more frequent use of lower-limb ultrasonography at earlier time points. S7 Fig shows a comparison of the number of days from the initial DVT diagnosis to the first post-diagnosis ultrasonography evaluation annually. The median interval was notably shortened in 2019 and 2020, suggesting that earlier follow-up may have contributed to the increased detection of asymptomatic VTE.

However, the exact reason for these shorter follow-up intervals remains unclear and is a limitation of our retrospective, single-center study. Changes in hospital policies or practices among physicians or departments may have influenced the results. These findings have been added to the “Results” section of the revised manuscript and the “Limitations” section has been expanded accordingly.

We evaluated the year-by-year trends in VTE recurrence (S6 Fig). In the cancer group, the cumulative incidence of VTE recurrence varied significantly by year, with notably higher recurrence rates observed in the latter part of the study period (2019–2020). In contrast, no significant differences in VTE recurrence by year were observed in the non-cancer group. To explore the factors contributing to these annual differences within the cancer group, we conducted Kaplan–Meier analyses, stratifying cases into symptomatic and asymptomatic VTE recurrences (S7 Fig). The results showed no significant variation by year in symptomatic VTE recurrence (S7A Fig); however, significant differences were noted in asymptomatic VTE recurrence, with a marked increase from 2019 to 2020 (S7B Fig). The precise reason for the increase in asymptomatic VTE recurrence during the latter part of the study period remains unclear. One plausible explanation is the change in follow-up practices after DVT diagnosis, leading to more frequent use of lower limb ultrasonography at earlier time points. S8 Fig compares the number of days from the initial DVT diagnosis to the first post-diagnosis ultrasonography, which was evaluated annually. The median interval was notably shortened in 2019 and 2020, suggesting that earlier follow-up may have contributed to the increased detection of asymptomatic VTE in these years. (pages 15-16, lines 226-240)

Furthermore, owing to the retrospective design, follow-up practices after the initial DVT diagnosis were at the discretion of individual physicians, and no standardized protocol was employed, which may have influenced the outcomes of this study. (pages 30-31, lines 506-508)

We also evaluated the Kaplan–Meier curves for major bleeding by year and found no temporal differences between the groups (S11 Fig). We have added this information to the “Major Bleeding” section of the revised manuscript as follows:

No temporal differences were observed in major bleeding events between the groups (S11 Fig). (page 19, lines 286-287)

We additionally evaluated the Kaplan–Meier curves for overall survival by year and found no temporal differences in either group (S14 Fig). We have added this information to the “Overall Survival” section and “Supporting information” in the revised manuscript as follows:

We also evaluated year-by-year trends in OS and found no temporal differences in either group (S14 Fig). (page 21, lines 333-334)

Major comment #2:

The authors do not analyse the role of cancer stage or treatment type (e.g., chemotherapy vs. radiation) in VTE recurrence or bleeding risks. These factors likely contribute to the observed differences in outcomes.

Reply:

We agree that the cancer stage and treatment type are important factors that may influence the risk of VTE recurrence and bleeding. Unfortunately, our dataset did not include information on cancer stage; however, it did include data on the presence or absence of distant metastases at the time of DVT diagnosis. We further stratified the patients into three groups—non-cancer, cancer without distant metastasis, and cancer with distant metastasis—to explore the differences in clinical outcomes. Kaplan–Meier curves were constructed to compare the cumulative incidence of VTE recurrence and major bleeding among the three groups. We found that VTE recurrence was lowest in the non-cancer group, with no significant differences between the cancer with and without metastasis groups (S5 Fig). However, major bleeding events occurred most frequently in patients with cancer and distant metastasis, followed by patients with cancer without metastasis and patients without cancer (S12 Fig). We have added this information to the “Results” section as follows:

Among the patients with cancer, 33.9% had distant metastases. (page 12, line 191)

We further stratified the patients into three groups—non-cancer, cancer without distant metastasis, and cancer with distant metastasis—to explore the differences in clinical outcomes. We found that VTE recurrence was lowest in the non-cancer group, with no significant differences between the cancer groups with and without metastasis (S5 Fig). (page 15, lines 222-226)

We further stratified the patients into three groups—non-cancer, cancer without distant metastasis, and cancer with distant metastasis—to explore differences in clinical outcomes. Major bleeding events occurred most frequently in patients with cancer with distant metastasis, followed by patients with cancer without metastasis and patients without cancer (S12 Fig). (page 19, lines 287-291)

Unfortunately, our registry did not comprehensively capture specific treatment types (e.g., chemotherapy or radiotherapy). Consequently, stratified or adjusted analyses based on these variables could not be performed. We have added this limitation to the “Limitations” section, as follows:

Finally, although we collected information on cancer type and the presence or absence of distant metastasis at baseline, we did not have access to detailed information regarding cancer stage or treatment. Consequently, we were unable to assess the influence of these factors on clinical outcomes. (page 31, lines 517-520)

Major comment #3:

The subgroup analyses of cancer type lack adjustments for multiple comparisons. This undermines the reliability of some findings, such as the higher recurrence risk in CNS cancer patients.

Reply:

As you correctly pointed out, the initial subgroup analyses by cancer type were not adjusted for multiple comparisons, which may have increased the risk of type I errors. In response, we reanalyzed the data using multivariate Fine-Gray models to adjust for potential confounders. For VTE recurrence, the model was adjusted for age, sex, body mass index, comorbidities as a whole, baseline pulmonary embolism, hemoglobin level, anticoagulation beyond the acute phase, and discontinuation of anticoagulants (S9 Fig). For major bleeding, the model was similarly adjusted, excluding the discontinuation of anticoagulants (S13 Fig). Based on these analyses, the descriptions in the revised manuscript have been updated as follows:

Further stratification by cancer type revealed that hematological malignancies, head/neck, cervical/uterine, ovarian/peritoneal, pancreatic, colorectal, and CNS cancers were significantly associated with a higher risk of VTE recurrence (S9 Fig). (page 18, lines 266-268)

Further stratification according to cancer type revealed that pancreatic, cervical/uterine, colorectal, kidney/ureter/bladder, and ovarian/peritoneal cancers were associated with a higher risk of major bleeding (S13 Fig). (page 21, lines 318-320)

Additionally, details of the analytical methods and covariates are provided in the “Statistical Analysis” section, as follows:

Given the potential impact of mortality as a competing risk, the Fine–Gray subdistribution hazard model was applied to estimate the subdistribution hazard ratios (sHRs) for VTE recurrence and major bleeding. In these models, death from other causes was treated as a competing event, whereas in the Cox model, it was treated as a censored event for comparison. Multivariate models were adjusted for age, sex, BMI, comorbidities as a whole, baseline PE, hemoglobin level, anticoagulation beyond the acute phase, and discontinuation of anticoagulant therapy. Notably, discontinuation was excluded from the adjustment in the major bleeding model because bleeding events occurring after discontinuation were not systematically captured in our registry. Subgroup analyses for VTE recurrence and major bleeding were conducted using Fine–Gray subdistribution hazard models, adjusting for the same covariates in the primary analyses. To evaluate the impact of VTE recurrence and major bleeding on OS, additional subgroup analyses were performed using Cox proportional hazards models, incorporating VTE recurrence or major bleeding as covariates along with the previously described variables. (page 9, lines 144-157)

Major comment #4:

The survival analyses focus primarily on differences between cancer and non-cancer groups but do not explore predictors of survival within the cancer group. For example, how does the presence of major bleeding or VTE recurrence impact survival in different cancer subtypes?

Reply:

We conducted multivariate analyses using the Cox proportional hazards model to evaluate the impact of VTE recurrence and major bleeding on overall survival in patients with and without cancer (S16 Fig). Among the cancer group, major bleeding was significantly associated with poorer survival (adjusted HR: 1.71; 95% CI: 1.33–2.19; P < 0.001), whereas VTE recurrence was not (adjusted HR: 1.06; 95% CI: 0.80–1.40; P = 0.696). In contrast, in the non-cancer group, both VTE recurrence (adjusted HR: 2.42; 95% CI: 1.16–5.05; P = 0.020) and major bleeding (adjusted HR: 5.24; 95% CI: 2.72–10.1; P = 0.043) were significantly associated with worse survival. Furthermore, we performed stratified analyses according to cancer type (S17 Fig). We found that VTE recurrence was significantly associated with decreased survival in ovarian/peritoneal cancer (adjusted HR: 2.91; 95% CI: 1.52–5.59; P < 0.001), while major bleeding was associated with poorer survival in kidney/ureter/bladder cancer (adjusted HR: 2.77; 95% CI: 1.07–7.19; P = 0.037), and upper gastrointestinal tract cancers (adjusted HR: 3.88; 95% CI: 1.08–13.9; P = 0.039). These results have been included in the "Overall survival" subsection of the Results section as follows:

We further investigated the impact of VTE recurrence and major bleeding on the OS of patients with and without cancer (S16 Fig). Among the cancer group, major bleeding was significantly associated with poorer survival (adjusted HR: 1.71; 95% CI: 1.33–2.19; P < 0.001), whereas VTE recurrence was not (adjusted HR: 1.06; 95% CI: 0.80–1.40; P = 0.696). In contrast, in the non-cancer group, both VTE recurrence (adjusted HR: 2.42; 95% CI: 1.16–5.05; P = 0.020), and major bleeding (adjusted HR: 5.24; 95% CI: 2.72–10.1; P = 0.043) were significantly associated with worse survival. Furthermore, we performed stratified analyses according to the cancer type. VTE recurrence was significantly associated with decreased survival in ovarian/peritoneal cancer (adjusted HR: 2.91; 95% CI: 1.52–5.59; P < 0.001), and major bleeding was strongly associated with poorer survival in kidney/ureter/bladder cancer (adjusted HR: 2.77; 95% CI: 1.07–7.19; P = 0.037), and upper gastrointestinal tract cancer (adjusted HR: 3.88; 95% CI: 1.08–13.9; P = 0.039) (S17 Fig). (pages 22-23, lines 346-357)

Specific comment #1:

In the abstract, details on subgroup findings or the implications of anticoagulation discontinuation, which are important outcomes, are lacking. Please include.

Reply:

As suggested, we have revised the Abstract to include the impact of anticoagulation discontinuation on VTE recurrence, which is one of the key findings of our study, as follows:

In the cancer group, discontinuation of bleeding-related anticoagulation therapy was associated with increased VTE recurrence (P < 0.001), whereas no such association was observed in the non-cancer group (P = 0.716). (page 3, lines 38-41)

Specific comment #2:

The introduction does not clearly differentiate how the study builds upon existing literature, such as the COMMAND VTE Registry studies. The rationale for focusing on cancer patients with DVT diagnosed via ultrasound is not sufficiently justified. A discussion of why asymptomatic cases are important to study would provide more clarity.

Reply:

As suggested, we have revised the Introduction section to clarify how our study differs from previous reports, such as the COMMAND VTE Registry (Sakamoto J, et al. Circ J 2019, PMID: 31548438) and COMMAND VTE Registry-2 (Chatani R, et al. Thromb Res 2024, PMID: 38190788), which primarily included patients with symptomatic VTE. Our study included both symptomatic and asymptomatic VTE cases, reflecting real-world cancer care. We have also cited recent studies suggesting that asymptomatic VTE may be associated with recurrence and poor prognosis in patients with cancer. This revision has been added to the “Introduction” section as follows:

In particular, existing studies, such as the COMMAND VTE Registry and COMMAND VTE Registry-2, primarily focused on symptomatic VTE and did not include asympt

---

## [Editor Report · Decision Letter 1]

Risk of recurrence and bleeding in patients with cancer-associated venous thromboembolism in the direct oral anticoagulants era: Findings from the TULIPE registry

PONE-D-25-11278R1

Dear Dr. Tajiri,

We’re pleased to inform you that your manuscript has been judged scientifically suitable for publication and will be formally accepted for publication once it meets all outstanding technical requirements.

Kind regards,

Sonu Bhaskar, MD PhD

Academic Editor

PLOS ONE

Additional Editor Comments (optional):

Thank you for submitting a revised version of your manuscript.

I am pleased to accept the manuscript in its current form. Thank you for submitting your work to PLOS One.
---

## [Editor Report · Acceptance letter]

PONE-D-25-11278R1

PLOS ONE

Dear Dr. Tajiri,

I'm pleased to inform you that your manuscript has been deemed suitable for publication in PLOS ONE. Congratulations! Your manuscript is now being handed over to our production team.

Kind regards,

on behalf of

Dr. Sonu Bhaskar

Academic Editor

PLOS ONE